

# Improvement of nuclear facilities DNP3 protocol data transmission security using super encryption BRC4 in SCADA systems

Eko Hadiyono Riyadi[1,2], Agfianto Eko Putra[1] and Tri Kuntoro Priyambodo[1]

[1] Department of Computer Science and Electronics, Universitas Gadjah Mada, Yogyakarta, DIY, Indonesia
[2] Centre for Regulatory Assessment of Nuclear Installations and Materials, Jakarta, Indonesia

## ABSTRACT

**Background**. Data transmissions using the DNP3 protocol over the internet in SCADA systems are vulnerable to interruption, interception, fabrication, and modification through man-in-the-middle (MITM) attacks. This research aims to improve the security of DNP3 data transmissions and protect them from MITM attacks.

**Methods**. This research describes a proposed new method of improving DNP3 security by introducing BRC4 encryption. This combines Beaufort encryption, in which plain text is encrypted by applying a poly-alphabetic substitution code based on the Beaufort table by subtracting keys in plain text, and RC4 encryption, a stream cipher with a variable-length key algorithm. This research contributes to improving the security of data transmission and accelerating key generation.

**Results**. Tests are carried out by key space analysis, correlation coefficient analysis, information entropy analysis, visual analysis, and time complexity analysis. The results show that to secure encryption processes from brute force attacks, a key of at least 16 characters is necessary. IL data correlation values were IL1 = −0.010, IL2 = 0.006, and IL3 = 0.001, respectively, indicating that the proposed method (BRC4) is better than the Beaufort or RC4 methods in isolation. Meanwhile, the information entropy values from IL data are IL1 = 7.84, IL2 = 7.98, and IL3 = 7.99, respectively, likewise indicating that the proposed method is better than the Beaufort or RC4 methods in isolation. Both results also show that the proposed method is secure from MITM attacks. Visual analysis, using a histogram, shows that ciphertext is more significantly distributed than plaintext, and thus secure from MITM attacks. The time complexity analysis results show that the proposed method algorithm is categorized as linear complexity.

# INTRODUCTION

Supervisory control and data acquisition (SCADA) is a control system architecture comprising computers, networked data communications, and graphical user interfaces (GUI) for high-level process supervisory management, as well as other peripheral devices

Corresponding author
Tri Kuntoro Priyambodo,
mastri@ugm.ac.id

such as programmable logic controllers (PLC) and discrete proportional–integral–derivative (PID) controllers, which are used to interface with machinery or process plants.

SCADA allows operators to change setpoint data from a distance, monitor processes, and obtain measurement information. It consists of three components: a remote terminal unit (RTU) to collect data from the sensor and remote device, a master terminal unit (MTU) equipped with a Human Machine Interface (HMI) for monitoring and control, and communication infrastructure to connect components (*Jain & Tripathi, 2013*; *Czechowski, Wicher & Wiecha, 2015*; *Shabani & Ahmed, 2014*). SCADA requires an industrial network protocol—a real-time communication protocol made to connect interface communication systems and instruments—to communicate with controlled devices.

In SCADA, security is an important factor (*Rezai, Keshavarzi & Moravej, 2017*), especially in critical industrial infrastructure. SCADA security systems connected to the internet are closely related to cybersecurity, and thus require special attention in critical industries. The development of cyberinfrastructure can improve the interconnection and security of smart networks (*Mathew, 2019*); indeed, several designs have sought to increase their investigative abilities (*Prayudi, Ashari & Priyambodo, 2015*).

Cyberattacks have damaged critical facilities, including nuclear facilities (*Kesler, 2011*). One example of protocol implementation is found in nuclear power plants (NPP), smart grid electricity facilities (*Nguyen, Ali & Yue, 2017*; *Orojloo & Azgomi, 2017*; *Dragomir et al., 2016*) that are strictly regulated owing to safety and security considerations. It is thereby necessary to ensure the safety and security of SCADA implementation (*Chen et al., 2015*; *Cremers, Dehnel-wild & Milner, 2017*). Several NPPs have been found vulnerable to cyberattacks, and various attempts have been made to improve nuclear security, using systematic (*Nguyen, Ali & Yue, 2017*) and dynamic mapping systems to determine which assets are most vulnerable to cyberattacks (*Orojloo & Azgomi, 2017*).

Famous industrial network protocols include Modbus (*Chen et al., 2015*), ICCP/-TASE.2 (*Robinson et al., 1995*), Distributed Network Protocol/DNP3 (*Cremers, Dehnel-wild & Milner, 2017*; *Bratus et al., 2016*; *Siddavatam & Kazi, 2016*), and OPC (*Dahal, Ujjwal & Cheten, 2011*). Each has its unique characteristics, as well as its own unique methods for verifying the integrity and data security. The specific requirements of industrial networks often make protocols particularly vulnerable to interference (*Czechowski, Wicher & Wiecha, 2015*; *Siddavatam & Kazi, 2016*; *Kapil et al., 2020*). Among the above-mentioned standard network protocols, DNP3 is the most popular.

Initially, DNP3 was designed for local network communication between MTU and RTU, or between RTU and IED. As most users implement DNP3 to communicate serially, the protocol was developed to work through routable protocols such as TCP/IP (*Shin, Eom & Song, 2015*). As a message protocol, DNP3 was developed to work over IP, thus making RTU communication more accessible *via* modem networks (*Kim et al., 2018*). The advantages of DNP3 over other protocols include its reliability, efficiency, and real-time transference of data, as well as its implementation of several standard data formats and support for data synchronization (both of which make real-time transmission more

efficient and reliable) (*Jain & Tripathi, 2013*; *Faisal, Cardenas & Wool, 2017*; *Senthivel, Ahmed & Roussev, 2017*).

However, the connection of SCADA systems to the internet network through the DNP3 protocol is also problematic, as its connections are potentially open to vulnerability loopholes. Such vulnerability can be used by attackers to steal the transmitted data. Furthermore, attackers may interrupt, intercept, fabricate, and modify the data, which would also hamper SCADA (*Nguyen, Ali & Yue, 2017*; *Orojloo & Azgomi, 2017*; *Dragomir et al., 2016*; *Senthivel, Ahmed & Roussev, 2017*; *Mantere, Sailio & Noponen, 2013*; *Lee et al., 2014*; *Park, Suh & Park, 2016*; *Bartman & Carson, 2016*; *Shitharth & Winston, 2016*; *Rahman, Jakaria & Al-Shaer, 2016*; *Sembiring, 2017*; *Amoah, Camtepe & Foo, 2016*).

It may therefore be concluded that data transmissions through DNP3 protocols in SCADA systems within critical industries are vulnerable to man-in-the-middle (MITM), brute force, eavesdropping, etc. This research aims to improve the security of plain data transmission through DNP3 protocols from the above-mentioned attacks.

## RELATED WORKS

Although SCADA is widely used due to its rapid development, recent studies have highlighted its vulnerabilities from a cyber-security and cyber-physical security perspective (*Bratus et al., 2016*; *Siddavatam & Kazi, 2016*; *Mantere, Sailio & Noponen, 2013*; *Lee et al., 2014*; *Rahman, Jakaria & Al-Shaer, 2016*; *Sembiring, 2017*; *Smaiah, Khellaf & Cherifi, 2016*; *Friesen, 2016*; *Moreira et al., 2016*; *Darwish, Igbe & Saadawi 2015*; *Amoah, 2016*; *Chen et al., 2014*; *Nivethan & Papa, 2016*; *Singh, Nivangune & Mrinal, 2016*). The threat of cyberattacks looms over SCADA systems that communicate with network protocols (*Mantere, Sailio & Noponen, 2013*; *Lee et al., 2014*; *Rahman, Jakaria & Al-Shaer, 2016*; *Amoah, 2016*; *Ahn et al., 2015*). As such, *Hou et al. (2016)*. investigated the detection of attacks within SCADA systems and network protocols, including the possibility of detecting attacks in smart grids using Dirichlet. Similarly, *Mantere, Sailio & Noponen (2013)* analyzed the detection of anomalies that could breach security.

The analysis and detection of network anomalies must be carried out continuously and periodically, perhaps through modeling and simulation using the OPNET Modeler method (*Hou et al., 2016*). Sniffing and DDOS attacks may be detected in smart grids with Novel IDS technology (*Shitharth & Winston, 2016*). A simulation test was conducted using NS-2 in the Novell IEEE802.15.4 protocol, finding that security performance improved between 95.5 and 97% (*Shabani & Ahmed, 2014*).

Several studies related to the security of DNP3 protocols in SCADA systems have also been conducted. Such studies have investigated how security systems can be implemented, tested, or developed in SCADA systems' DNP3 protocol. One such study tested communication protocol using DNPSec (*Jain & Tripathi, 2013*). Shahzad, meanwhile, tested three layers of DNP3 using dynamic cryptographic buffers, showing that it could reduce the effectiveness of attacks and improve security (*Shahzad, Musa & Irfan, 2014*). Research about the application of security authentication using a Tamarin model in a smart grid, meanwhile, showed that a DNP3-SA protocol meets security standards (*Cremers, Dehnel-wild & Milner, 2017*).

The testing of the DNP3 protocol was also proposed by developing a Linux-based firewall for industrial control systems, especially in the power sector, using the U32 byte matching feature (*Nivethan & Papa, 2016*). Research related to broadcast security messages from MTU to several other stations in DNP3-SAB protocol in SCADA systems showed that broadcasts can be secured from several attack vectors, such as modification, injection, spoofing, and replay (*Amoah, 2016*). A vulnerability analysis of the DNP3 protocol was carried out by observing specific surface attacks within the Data Link and Application layers (*Singh, Nivangune & Mrinal, 2016*).

Elsewhere, DNP3 safety analysis was conducted using a Colored Petri Nets model and Linear Discriminant Analysis, finding that the system can successfully detect and reduce abnormal activity (*Tare et al., 2016*). The development of a security assessment framework for cyber-physical systems was conducted by investigating the DNP3 protocol's resistance to attacks, including passive network monitoring, response replay requirements, rogue interoperations, buffer flooding, and TCP veto (*Siddavatam & Kazi, 2016*). Through vulnerability and penetration testing, MITM attacks were modeled using grid technology to evaluate cybersecurity threats to SCADA systems implementing DNP3 protocols (*El Bouanani et al., 2019*). Two attack scenarios were used: an unsolicited message attack and injection data collection (*Darwish, Igbe & Saadawi, 2016*).

Cryptography aims to ensure that the data sent is correct and accessed by the right people. Research on secure network protocols was carried out by applying encryption using the Diffie-Hellman method (*Isa et al., 2014*). *Premnath, Jo & Kim (2014)*, testing the NTRU cryptographic method, found that it has a faster runtime than RSA at the same security level. Testing of new encryption methods was also carried out using an i-key to implement a secure communication system (*Atighehchi et al., 2010*). The use of an i-key as a cryptographic protocol during dynamic re-locking was carried out by testing and simulating the IEEE802.1x standard protocol using WEP system encryption (*Yu & Pooch, 2009*). It is claimed that such research might prevent MITM attacks because the decryption processes can only be executed by authorized senders and recipients, *i.e.,* those who update the i-key.

The security of the DNP3 protocol in SCADA was improved using the bump-in-the-wire method, which consists of key distribution, cryptography, and intrusion detection (*Mohd, Singh & Bhadauria, 2019*). Using IDS, Jain tested a combination of syntactic and semantic detection techniques, with the Diffie-Helman method used for locking and DNPSec used as communication protocol. Testing showed that DNP3 security can be improved through efficient management and crypto-key distribution, while DNPSec can identify other packets on the DNP3 network (*Jain & Tripathi, 2013*).

Furthermore, research into dynamic cryptographic buffers was conducted using eight remote units divided into two stations (S-bed1) and sixteen isolated groups divided into two stations (S-bed2). The study was successful in preventing an MITM attack (*Shahzad et al., 2016*). Communication between MTU and RTU was subsequently verified using openDNP3, with Scapy equipment used for packet manipulation and penetration testing. The results showed that OpenDNP3 v1.1.0 supports the prevention of attacks that

affect confidentiality, such as MITM (*Singh et al., 2015a*). Protecting the integrity and confidentiality of data is important for any network protocol (*Fremantle & Scott, 2017*).

Elliptic cryptography and hash functions have been developed to analyze performance, as has the 3PAKE protocol, and third-party key exchange authentication. Tests were carried out using AVISPA simulator software, showing that the protocol can efficiently prevent active or passive attacks (*Islam et al., 2017*). *Dey & Ghosh (2018)* analyzed the crypto security of four-bit and eight-bit crypto S-boxes, finding that S-box crypto security is better than DES and AES.

## Comparison to other hybrid cipher approaches

To prevent cyberattacks, any data communication system must include strong data transmission security, perhaps using cryptography (*Mohamed et al., 2020*). Every cryptographic scheme has its own strengths and weaknesses, and thus the application of a single cryptographic technique has severe shortcomings. To secure data, without compromising security, a cost-effective symmetric encryption method is often used. However, in such processes, proper key distribution is problematic (*Riyadi, Priyambodo & Putra, 2021*). An asymmetric scheme has potential. Unfortunately, however, the process is slower and consumes more computer resources than symmetric encryption. The integration of several cryptographic methods is therefore proposed to provide efficient data security while simultaneously addressing the problem of key distribution, thereby overcoming each scheme's security weaknesses (*Riyadi, Priyambodo & Putra, 2021*).

Some previous studies have utilized asymmetric cryptography; for example, *Purevjav, Kim & Lee (2016)* and *Harba (2017)* employed Rivest Shamir Adleman (RSA), while *Hong & Xuefeng (2013)* and *Xin (2015)* used Elliptic Curve Cryptography (ECC). Other studies have applied symmetric cryptography, *i.e., Altigani & Barry (2013), D'souza & Panchal (2017), Xin (2015)*, and *Harba (2017)* used Advanced Encryption Standard (AES), while *Hong et al. (2017)* implemented the Data Encryption Standard (DES) in combination with Rivest Code 4 (RC4). *Singh et al. (2015b)* similarly used symmetric encoding, while *Purevjav, Kim & Lee, (2016)* combined a public key encryption system with a symmetric hash function, thereby ensuring that messages encrypted with the public key could only be decrypted reasonably quickly using the private key. *Harba (2017)* proposed a method of protecting data transfer using a hybrid technique: to ensure secure transmission, a symmetric AES algorithm was used to encrypt files; an asymmetric RSA algorithm was used to encrypt AES passwords; HMAC was used to encrypt passwords and symmetric data.

*Hong & Xuefeng (2013)* used the ECC password algorithm and the SM2 handshake agreement to solve security problems in the information transmission process, but failed to conduct a performance evaluation. *Xin (2015)* proposed a mixed approach to encryption, integrating MD5 with ECC and AES, but again failed to evaluate performance results. *Altigani & Barry (2013)* proposed combining AES with the Word Shift Coding Protocol steganography protocol, producing a model that improved the confidentiality of messages and overall system security. *D'souza & Panchal (2017)* proposed a hybrid approach, combining Dynamic Key Generation and Dynamic S-box Generation with

**Table 1  Hybrid security approach to data transmission.**

| Study | Securing data transmission | Methods | Performance measuring | Provides layered security | Provide security analysis |
|---|---|---|---|---|---|
| N. Hong (*Hong & Xuefeng, 2013*) | ✓ | Handshake agreement (SM2) and ECC. | No performance evaluation. | – | – |
| Altigani (*Altigani & Barry, 2013*) | ✓ | AES and steganography Word Shift Coding. | Encryption time and extraction time. | ✓ | – |
| Xin (*Xin, 2015*) | ✓ | MD5, AES and ECDH. | Key exchange time, number of time; key length, time of signature, number of signature, verification time. | ✓ | – |
| Singh (*Singh et al., 2015b*) | ✓ | Symmetric encipherment and middle value algorithm. | Encryption and decryption test | ✓ | – |
| Purevjav (*Purevjav, Kim & Lee, 2016*) | ✓ | Symmetric cipher Ping Pong-128, RSA and hash function MD5. | Encryption and decryption test. | ✓ | – |
| Z. Hong (*Hong et al., 2017*) | ✓ | DES and RC4. | No evaluation. | ✓ | – |
| Harba (*Harba, 2017*) | ✓ | AES, RSA and HMAC. | Ciphertext size, encryption time | ✓ | – |
| D'souza (*D'souza & Panchal, 2017*) | ✓ | AES and Dynamic Key Generation and Dynamic S-box Generation. | Encryption and decryption test. | ✓ | – |
| Proposed Method | ✓ | Super Encryption BRC4, Dynamic Symmetric Four-key-generation. | Keyspace analysis, Correlation coefficient analysis, Information Entropy analysis, Visual analysis, Time complexity analysis, Encryption and decryption test. | ✓ | ✓ |

an AES algorithm. This method used Dynamic Key Generation to add data complexity, thereby increasing confusion and diffusion in the ciphertext.

*Hong et al. (2017)* offered a hybrid crypto algorithm that used the DES and RC4 encryption algorithms to encrypt communication data, but did not perform a performance evaluation. *Singh et al. (2015b)* proposed a hybrid encryption scheme that made it difficult for attackers to learn information from messages sent through insecure data transmissions. Table 1 provides a comparison of several approaches to securing data transmission offered by previous studies, including the method provided by this study. All of these studies aim to secure transmission data using hybrid cryptography and generate multiple keys to increase security. Likewise, although almost all of these studies provide a layered or graded approach to security, few provided a security analysis.

## MATERIALS & METHODS

### The BRC4 super encryption model

This study introduces BRC4 (Beaufort RC4) super encryption, a combination of Beaufort and RC4 encryption. Beaufort encryption converts plain text to a poly-alphabetic

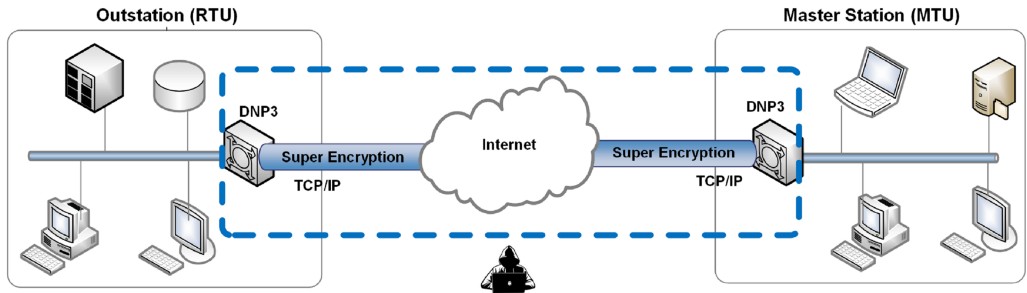

**Figure 1 The proposed research method.**

substitution code based on the Beaufort table, and although this algorithm is simple in its calculation processes, it still generates secure random numbers (*Sari & Hayati, 2018*). RC4, meanwhile, is a stream cipher with a variable-length key algorithm, which improves the confidentiality, randomness, and security of key streams (*Zhang, Liu & Ni, 2020*).

This research was conducted by simulating BRC4 super encryption on PLC program data (in the instruction list [IL] format) from industrial machines. This simulation is used to anticipate MITM attacks; in other words, BRC4 super encryption (a combination of Beaufort and RC4 encryption) is used to increase security and avoid MITM attacks (see Fig. 1).

The BRC4 super-encryption model consists of two models: an encryption model, *i.e.,* a combination of the Beaufort and the RC4 encryption processes (see Fig. 2), and a decryption model, *i.e.,* a variety of RC4 and Beaufort decryption processes (see Fig. 3). The encryption model is installed in the data transmission section (RTU), while the decryption model is established in the data receiving section (MTU). This simulation model is built and tested using hardware with the following specifications: i7-6500U processor, 16GB RAM, and Windows 10 operating system (64-bit).

## Encryption model design

Simulation is performed using Instruction List (IL) data from the Programmable Logic Controller (PLC) of several industrial machines. This IL data is in the form of an input-or-output logic command. Several IL data are used, with the following specifications: IL1 = 188 lines, IL2 = 881 lines, and IL3 = 4,571 lines. Each line consists of approximately 15 characters; as such, IL1 consists of 1,086 characters, IL2 consists of 7,158 characters, and IL3 consists of 33,046 characters.

### Data reading

The plaintext data are in the form of IL, *i.e.,* a PLC program containing lists of sequence logic instructions in the form of sequentially executed text. In this paper, the data used is in the IL1 format.

### Converting to one-line format

The plaintext data (IL1), presented in 188 lines and 1 column, is converted into a one-line array. To separate the lines, a semicolon (;) is used. This produces the following:

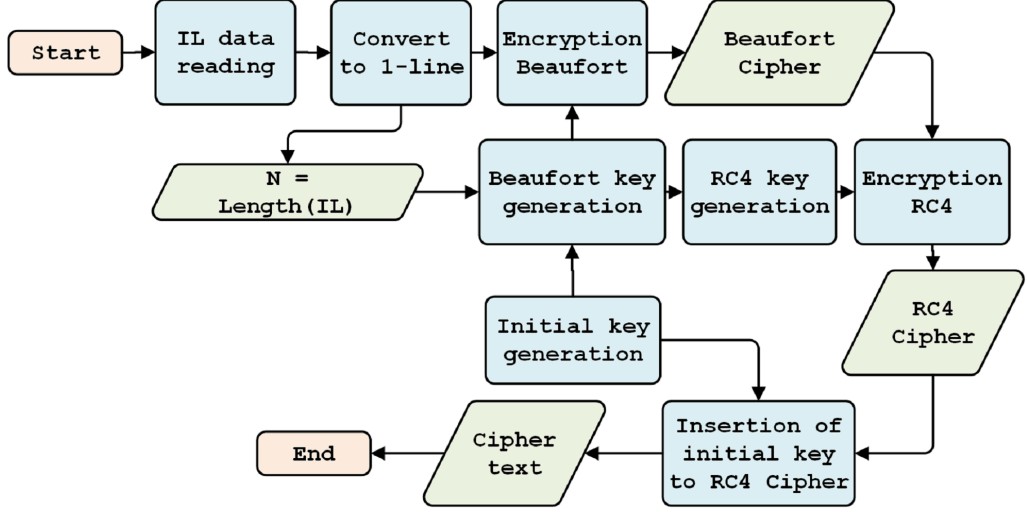

**Figure 2  Encryption model.**

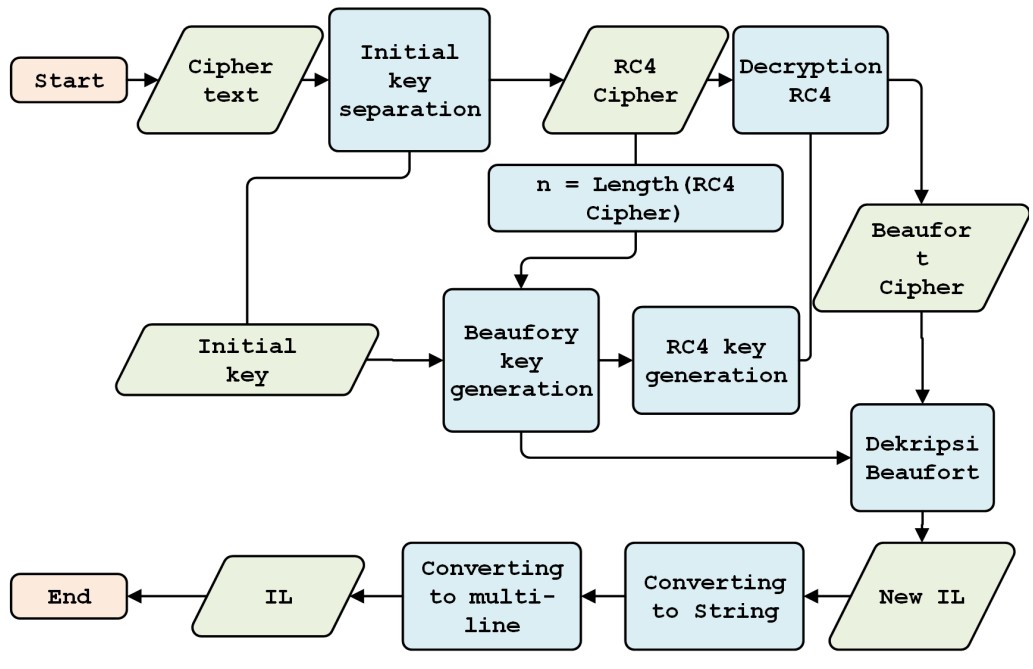

**Figure 3  Decryption model.**

LD M8002;SET S0;STL S0;ZRST S20 S80;RST M0;OUT T0 K1;LD T0;ANI X001;ANI X012;MPS;ANI X015;SET S20;MPP;AND X015;SET S25;SET S30;STL S20;LD X013;ANI X002;OUT Y006;LD X014;ANI X003;OUT Y007;LD X015;OR X001;SET S0;LD X012;SET S90;RST S20;STL S90;LD M8013; and so on.

Next, it is necessary to convert data from string format to numeric format to perform mathematical operations.

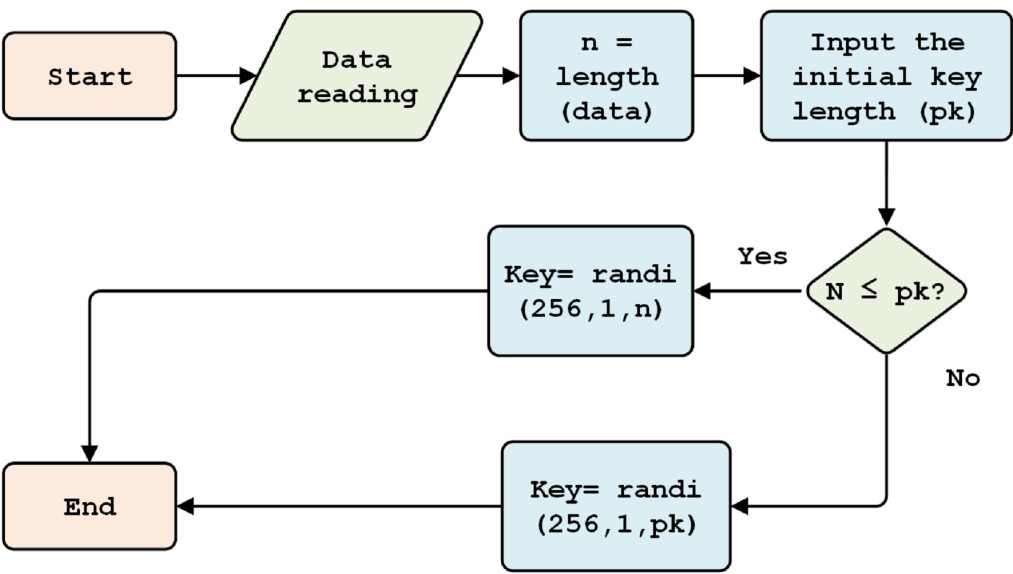

**Figure 4  Initial key generation.**

### Initial key generation

This process aims to generate a random initial key, one that cannot easily be detected by attackers. Random initial key generation is conducted during each data transmission process. Generation is carried out in the range of 1 to 256 bytes, with the length of the key depending on data size (*n*) and character length of the initial key (*pk, i.e.,* 64, 128, 256, 512, 1,024, 2,048 bits); the longer the initial key, the more secure.

Figure 4 shows that, if the length of the plaintext (*H*) is less than or equal to the size of the key, the length of the key will equal the length of the data (*n*). Otherwise, if the length of the data is greater than the length of the key, then the length of the key is equal to the length of the chosen initial key (*pk, 256*).

### Generation of Beaufort encryption key

In Beaufort key generation, if the initial key is shorter than plaintext, the initial key is generated repeatedly along with the plaintext. Such generation uses a keystream generator approach, following Eq. (1).

$$k_i = (k_{i-m} + k_{i-1}) \bmod 256 \tag{1}$$

$k_i$ is the key number-*i*, $k_{i-m}$ is the key number-i subtracted by the initial key (*m*), and key number-i subtracted by *1*. The initial key length is *256* characters. Since the plaintext is *1,086* characters long, a Beaufort key is generated to be equal in length to the plaintext. Key numbers *257* to *1,086* are generated based on Eq. (1). For example:

$k_{257} = (k_1 + k_{256}) \bmod 256$
$k_{257} = (206 + 174) \bmod 256$
$k_{257} = 380 \bmod 256$
$k_{257} = 124$

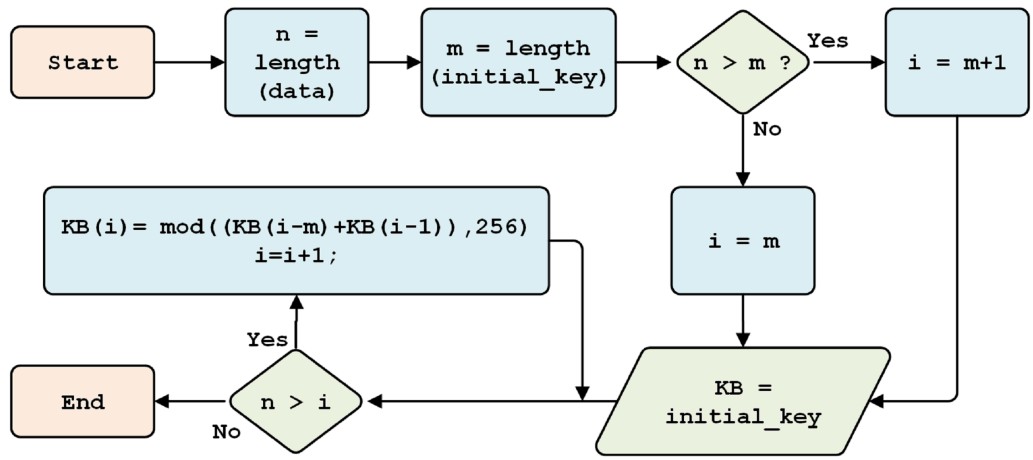

**Figure 5 Beaufort key generation.**

Figure 5 explains the process through which the Beaufort encryption key is generated. This process produces the Beaufort key for encryption, with the first *256* characters equal to the initial key and the generated results used for keys *257* to *1,086*.

### Beaufort encryption

Beaufort encryption uses the IL numeric format as plaintext and the Beaufort key. It is conducted using the following equation:

$$E_k\left(P_1, P_2, \ldots, P_m\right) = \left(K_1 - P_1, K_2 - P_2, \ldots, K_m - P_m\right) \; mod \; 256 \tag{2}$$

$$D_k\left(C_1, C_2, .., C_m\right) = \left(K_1 - C_1, K_2 - C_2, \ldots, K_m - C_m\right) \; mod \; 256 \tag{3}$$

where *E* (encryption), *D* (decryption), *P* (plain text), *C* (cipher), *K* (key). The encryption process begins with reading the plaintext (*n*), namely the IL numeric format. A subtraction operation is then carried out on each plaintext character, using a Beaufort key with a base 256 modulo, thereby forming a Beaufort cipher (CB) is formed, as shown in Fig. 6.

### Generation of RC4 encryption key

RC4 Key Generation consists of three steps. First, keystream generation is used to form the first cipher block. Second, permutation generation is conducted with a key scheduling algorithm function. Third, a pseudo-random number generation algorithm is used.

A key scheduling algorithm is used to initialize the permutations of array *S*. Key length is defined as the number of bytes contained within a key, that is, between *1* and *256*. The *S* array is initialized to the permutation identity; the *S* array is processed to 256 iterations. After retrieving the random *S* array, it is re-initialized, with the values of *i* and *j* being zero.

The PRGA process subsequently generates an RC4 key, by incrementing *i*, adding the values *S[i]* and *S[j]*, and swapping two values. An *S* value with an index equal to the numeric value *S[i]*, and *S[j]*, modulo 256, yields the RC4 key (see Fig. 7).

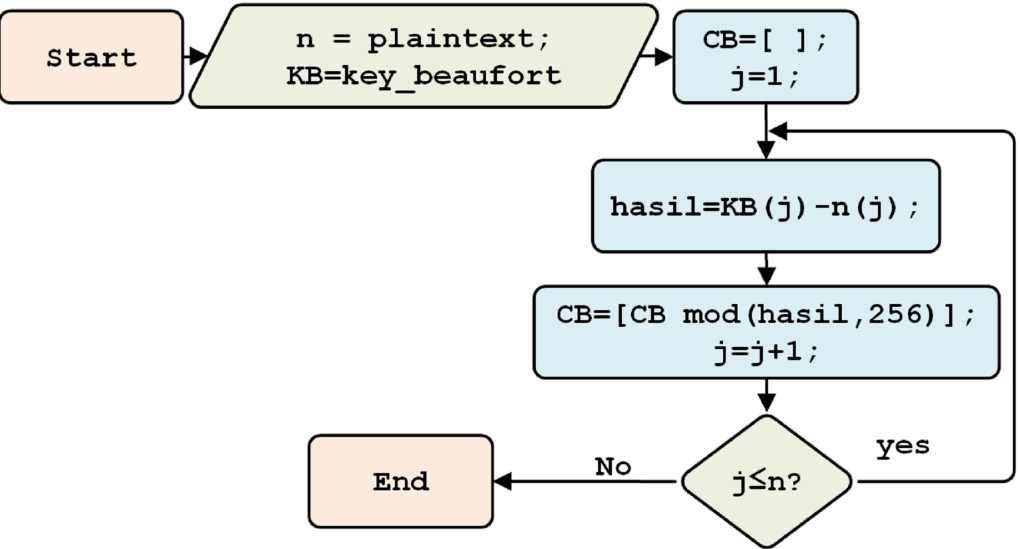

**Figure 6  The Beaufort encryption process.**

### RC4 encryption

RC4 encryption begins by reading the Beaufort cipher (CB) as plain text (Fig. 8). The system checks the plaintext ($n$); if $n < 256$, RC4 encryption is carried out against plain text n. Otherwise, blocks will be created using ($n/256$), rounded up and stored in STL. Subsequently, $n$ is checked again, if $n < 256$, the RC4 encryption process is conducted along with the plaintext characters ($n$). However, if $n > 256$, the permutation process in the next block forms the following array: $S[i]$ and $S[j]$. The values are then exchanged.

### The insertion of the initial key

This section describes the process of inserting the key and key information by including them behind the cipher through a combination of encryption processes. The key entered is the initial key, produced by random generation, and the key information is the length of the preselected key. The key and key information are used to perform decryption when the password data has been received. A randomly generated initial key is used to avoid MITM attacks.

While an attacker was to obtain ciphertext data, a combination of the cipher, initial key, and key information, said attacker would not obtain any information, as the ciphertext is random. If the attacker manages to separate the initial key from the ciphertext, the attacker will still not be able to read the cipher. The initial key is a different character length than the decryption key and can only be used after three generations: keystream generation, Beaufort decryption generation, and RC4 decryption generation.

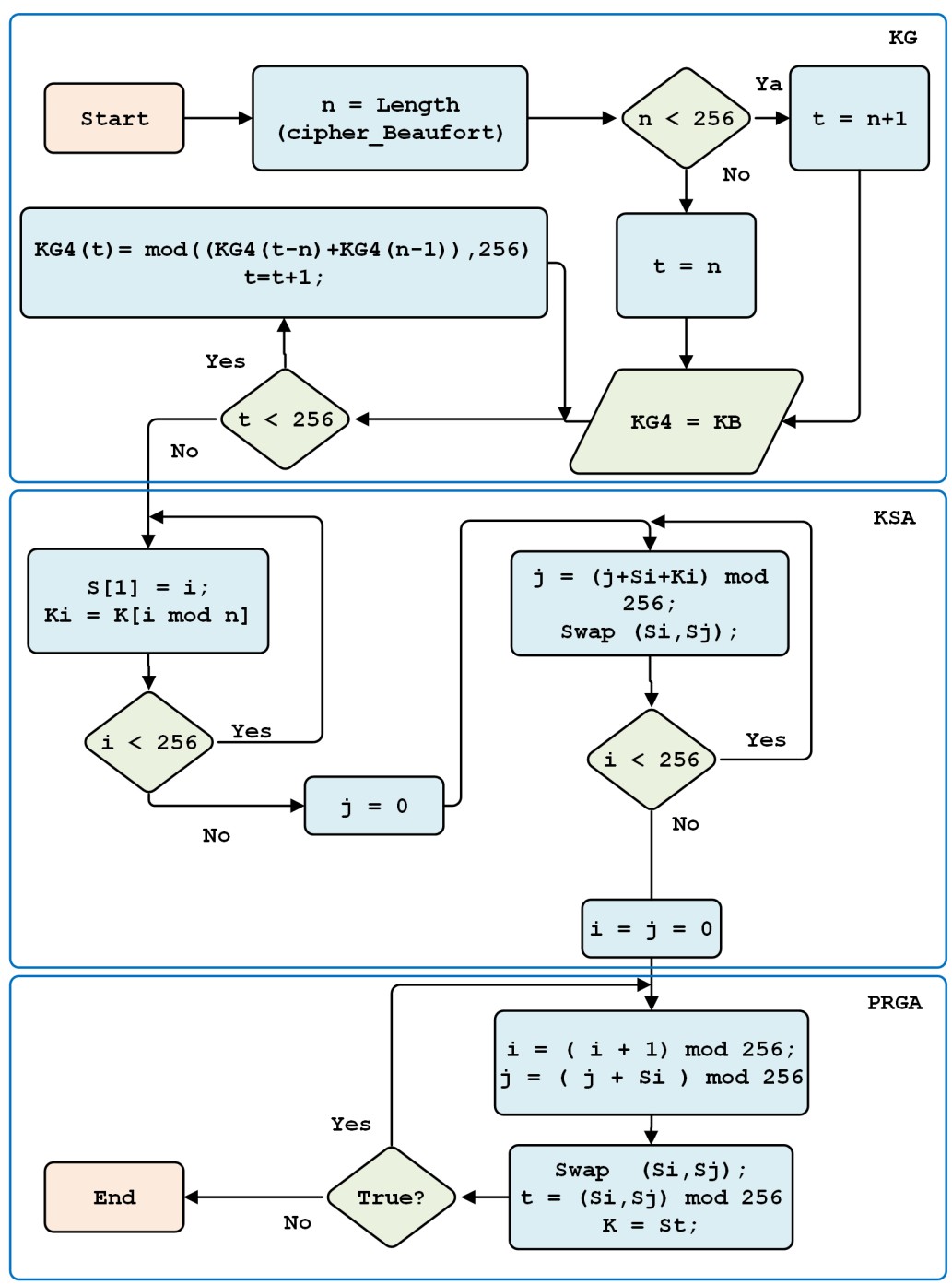

**Figure 7  RC4 key generation.**

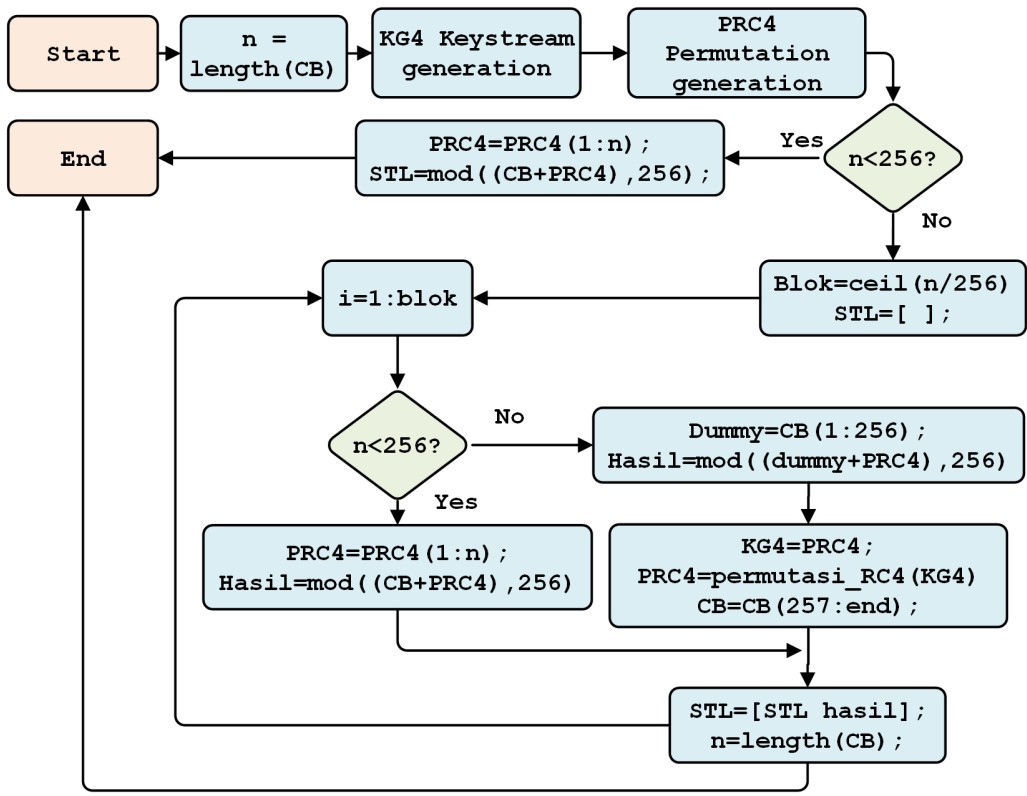

**Figure 8  The RC4 encryption process.**

## Decryption model design
### Initial Key Separation
After the ciphertext is received, it is separated into three components: the cipher, the initial key, and the key information (containing the key character length). The separation process is shown in Fig. 9.

### Beaufort decryption key generation
This section discusses the generation of Beaufort's decryption keys, which follow the process illustrated in Fig. 5. First, the system calculates the cipher data ($n$) and initial key ($m$). If $n$ is more than $m$, one is added to the variable $i$; otherwise, $m$ is kept in variable $i$, and the original key is saved as the Beaufort key. If the length of $n$ is greater than $i$, keystream generation is performed.

### RC4 decryption key generation
After generating the Beaufort decryption key, the RC4 key is produced through several stages: Keystream Generation (KG), Key Scheduling Algorithm (KSA) generation, and pseudo-random generation algorithm (PRGA).

Keystream generation aims to form the first block array and obtain a block length of up to 256 characters. A random block array key scheduling algorithm is generated based on the previous key (*i.e.,* the Beaufort decryption key). Finally, the RC4 decryption key is

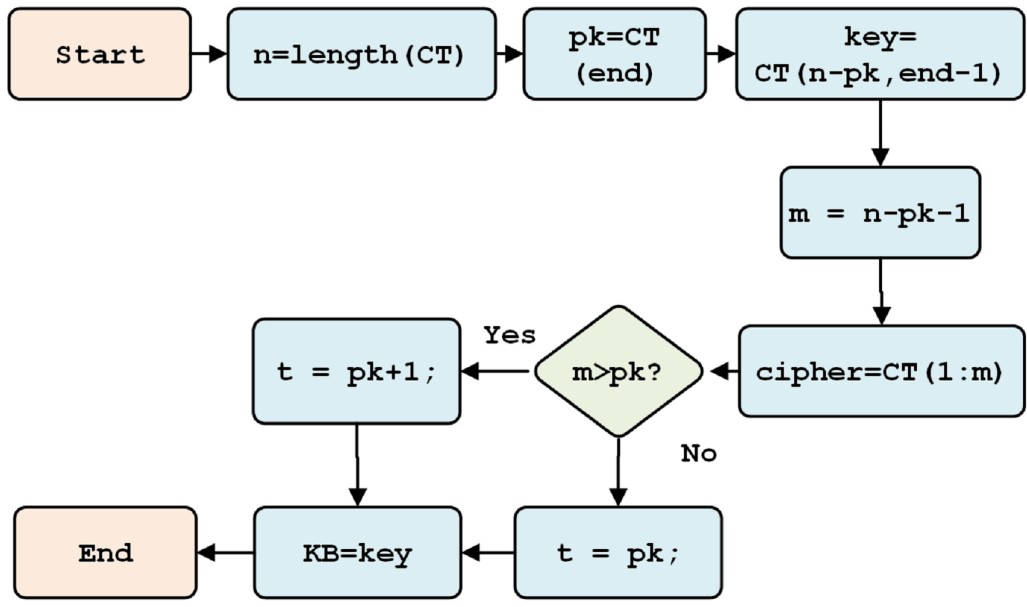

**Figure 9  Initial key separation.**

obtained; this key is used for the final decryption process. In its process, the generation of the RC4 decryption key resembles the generation of the RC4 encryption key. This approach is described in Fig. 7.

### RC4 Decryption

RC4 decryption is conducted, using the RC4 key, to separate the cipher data. The RC4 decryption process is described in Fig. 10. This process begins with the reading of the number of cipher characters. The subsequent processes through which keystreams and key scheduling algorithms are generated are intended to randomize the block array position. This is followed by generating a pseudo-random algorithm to obtain the RC4 key.

Therefore, if $n < 256$, the RC4 decryption process will produce the Beaufort cipher (CB). If $n > 256$, a block is formed to repeat the permutation of each character in the key array and S array, as well as to swap the values of $S[i]$ and $S[j]$. As a result, RC4 decryption (*i.e.,* the Beaufort cipher) is obtained.

### Beaufort decryption

This section describes Beaufort decryption, with the Beaufort cipher (CB) used as input, using the Beaufort Key (KB). The detailed process is shown in Fig. 11.

This process begins by calculating the length of the CB, as stored in variable $n$. The character length of the initial key is stored in variable $m$. If the length of the ciphertext is greater than the length of the initial key ($n>m$), the initial key is reproduced until the lengths are equal. The decryption process is done by adding each key character to CB, then creating an array to store the results of decryption. This process produces the plaintext (New_IL) in numeric format.

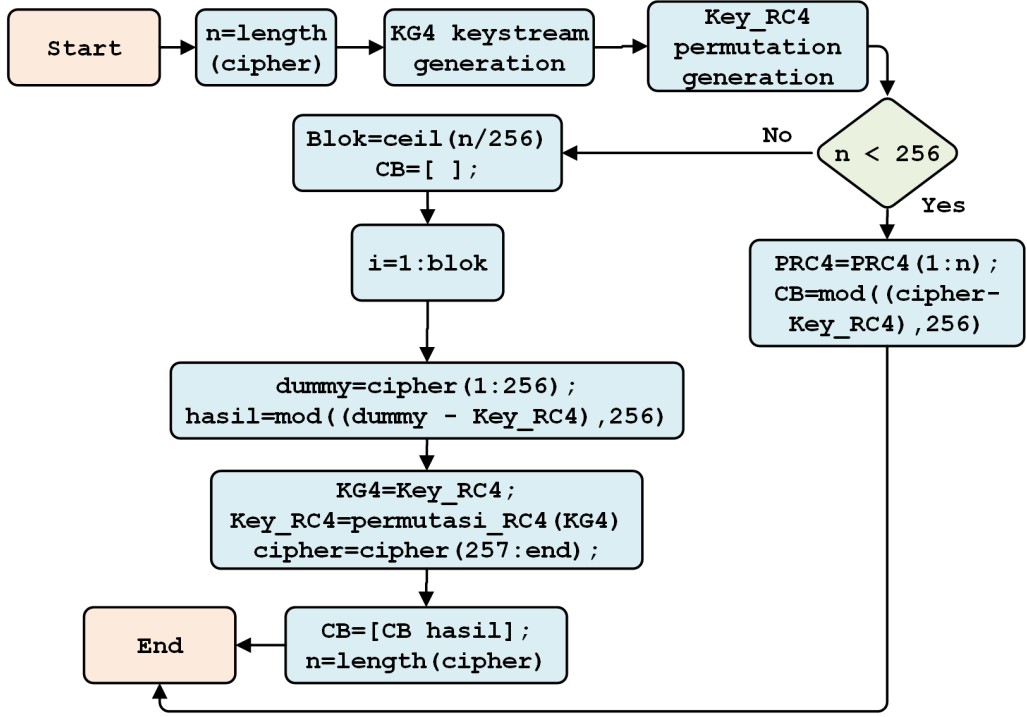

**Figure 10 The RC4 decryption process.**

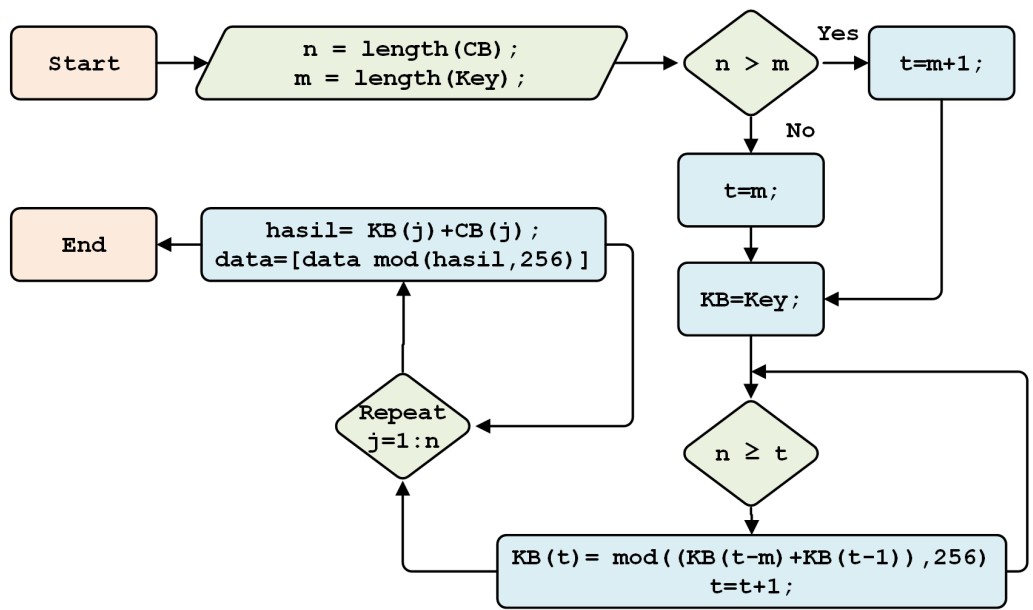

**Figure 11 The Beaufort decryption process.**

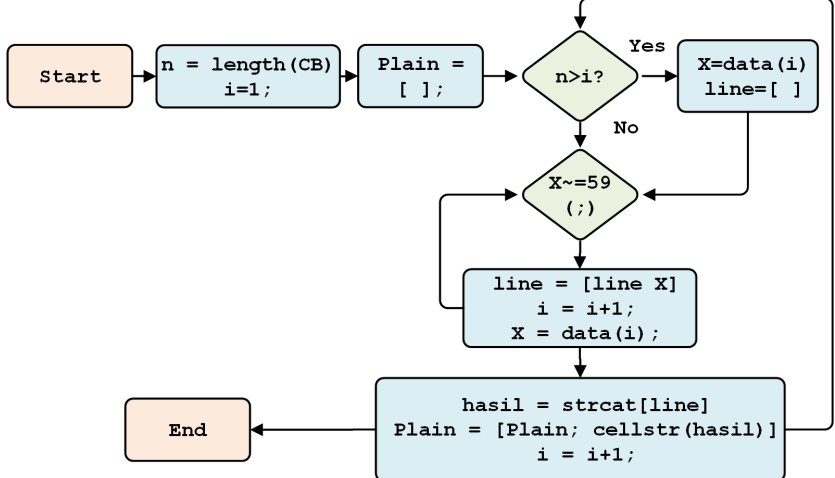

**Figure 12** The final model process.

### Design of the final model

This stage converts the numeric-type plaintext to the string type. It then changes the original one-line plain text format, and separator (;), into the original multi-line format. The process is detailed in Fig. 12.

## RESULTS AND DISCUSSION

This process reproduces IL data from the BRC4 super encryption system, which aims to increase the security of data transmission in SCADA systems *via* the DNP3 protocol. Its product is equivalent to the data sent *via* DNP3. As such, data can successfully be encrypted and decrypted using the BRC4 super-encryption method.

### Keyspace analysis

Keyspace analysis involves the analysis of keyspace within which the encryption system secures cipher data from brute force attacks. Brute-force attacks work by counting every possible combination that can form a password, and then testing them to determine the correct password. As the lengths and combinations of passwords grow, the amount of time it takes to find the correct password increases exponentially. Encryption systems must have very large keyspaces (greater than $2^{100}$ bits) to render brute force attacks ineffective (*Hamdi, Rhouma & Belghith, 2017*; *Setyaningsih, Wardoyo & Sari, 2020*).

The proposed model consists of several key generators. The initial key generator, randomly created for each session, has key lengths of 64, 128, 256, 512, 1,024, and 2,048 bits. Each initial key character is generated from 256 bytes ASCII code, and thus has a spread value of 1 to 256. A random initial key with a length of 16 characters (256 bits), using 256-byte ASCII code, would have a keyspace of $256^{16}$, equivalent to $2^{128}$. As such, an initial key that is 16 characters in length would be secure from brute force attacks, as would longer keys (*i.e.,* 256, 512, 1,024, and 2,048 bits), as shown in Table 2 (*Riyadi, Priyambodo & Putra, 2020*). Table 2 shows that a key 64 bits in length would not be secure from brute

**Table 2  Keyspace value (bits).**

| The key length | Keyspace value | Summary |
|---|---|---|
| 64 bits | $(2)^{64}$ | Not secure |
| 128 bits | $(2)^{128}$ | Secure |
| 256 bits | $(2)^{256}$ | Secure |
| 512 bits | $(2)^{512}$ | Secure |
| 1,024 bits | $(2)^{1024}$ | Secure |
| 2,048 bits | $(2)^{2048}$ | Secure |

**Table 3  Pearson correlation coefficient.**

| | | | | Coefficient of correlation (r) | | | | |
|---|---|---|---|---|---|---|---|---|
| Perfect | High | Moderate | Low | No correlation | Low | Moderate | High | Perfect |
| $-1$ | $\leq -0.90$ | $\leq -0.50$ | $\leq -0.30$ | $-0.29 \leq r \leq +0.29$ | $\geq +0.30$ | $\geq +0.50$ | $\geq +0.90$ | 1 |

**Table 4  Correlation coefficient value.**

| Encoded | IL1 | IL2 | IL3 |
|---|---|---|---|
| Beaufort only | $-0.046$ | 0.013 | 0.008 |
| RC4 only | $-0.093$ | $-0.126$ | 0.437 |
| Proposed method (BRC4) | $-0.010$ | 0.006 | 0.001 |

force attacks; a minimum key size of 128 bits is necessary to guarantee a secure encryption process.

## Correlation coefficient analysis

Correlation coefficient analysis aims to determine the correlation between plaintext and ciphertext data. If the correlation value is equal to 1, it means that the two data are the same.

Conversely, if the correlation value is lower than one (or close to 0), the two data are different; there is thus no relationship, and increased randomness (see Table 3). The less related the text, the better, as increased randomness means increased difficulty deciphering the relationship between plain text and encoded text (*Setyaningsih, Wardoyo & Sari, 2020*; *Setyaningsih & Wardoyo, 2017*). The correlation between plaintext and ciphertext data is formulated as:

$$r = \frac{n\left(\sum xy\right) - \left(\sum x\right)\left(\sum y\right)}{\sqrt{[n\sum x^2 - (\sum x)^2][n\sum y^2 - (\sum y)^2]}} \tag{4}$$

In this formula, $r$ is the correlation value, $x$ is the plaintext data, and $y$ is the ciphertext data. Based on Eq. (4), it can be seen that the proposed method produces a correlation value of $-0.010$ for IL1 data, 0.006 for IL2 data, and 0.001 for IL3 data (see Table 4).

Referring to Pearson (Table 3), the correlation value for all three data may be categorized as "no correlation," meaning that the plaintext and ciphertext data are not the same (*i.e.,* uncorrelated). Furthermore, the correlation value for the three data is closer to zero than

| Table 5 Information entropy value. | | | |
|---|---|---|---|
| Encoded | IL1 | IL2 | IL3 |
| Beaufort only | 7.77 | 7.91 | 7.94 |
| RC4 only | 7.69 | 7.39 | 5.49 |
| Proposed method (BRC4) | 7.84 | 7.98 | 7.99 |

the correlation value for data encrypted using the Beaufort cipher or RC4 in isolation. This shows that BRC4 super-encryption can improve the security of data transmission.

## Information entropy analysis

In cryptographic theory, information entropy is defined as a measure of the randomness of the amount of information in a message. Entropy is expressed in units of bits to express the degree of information randomness. Under random conditions, encrypted information with ciphertext data should have an optimum entropy value close to $\approx 8$; an entropy close to 8, thus, indicates that an encryption system is designed to be secure from MITM attacks (*Setyaningsih, Wardoyo & Sari, 2018*). The entropy value may be determined using the following equation (*Shukla & Kumar, 2016*):

$$H = -\sum_{k=0}^{n} P(k) Log_2(P(k)) \tag{5}$$

With H as the entropy value, n is the number of different symbols or codes in a message, and P(k) is the probability of symbol occurrence in the ciphertext.

Using the proposed method, entropy values of 7.84 (for IL1), 7.98 (for IL2), and 7.99 (for IL3) were returned. The proposed model's IL data has an entropy value close to 8, which indicates a high degree of randomness, and thus the ciphertext is secure from MITM attacks. A comparison of the entropy values for data encrypted using the Beaufort, RC4, and BRC4 models is provided in Table 5. Data encrypted through BRC4 super-encryption is closest to 8.00 in value, which indicates that BRC4 super-encryption is more secure than Beaufort or RC4 encryption alone.

## Visual analysis

Visual analysis aims to measure the results of IL data encryption using a histogram and compare the distribution of plaintext and ciphertext data. When a ciphertext histogram is more diverse and differently distributed than the plaintext histogram, it can be concluded that the ciphertext does not provide any clues or information that can be deciphered by MITM attacks.

Figure 13 shows the histogram results for IL data (the first 500 of 33,046 characters). It notes that data distribution between plaintext and ciphertext is very varied, and thus ciphertext data is secure from MITM attacks. The plaintext is distributed in the numeric range of 10 to 99, while the ciphertext has a distribution of 1 to 256. As such, the ciphertext is more secure from MITM attacks than texts encrypted using Beaufort or RC4 in isolation.

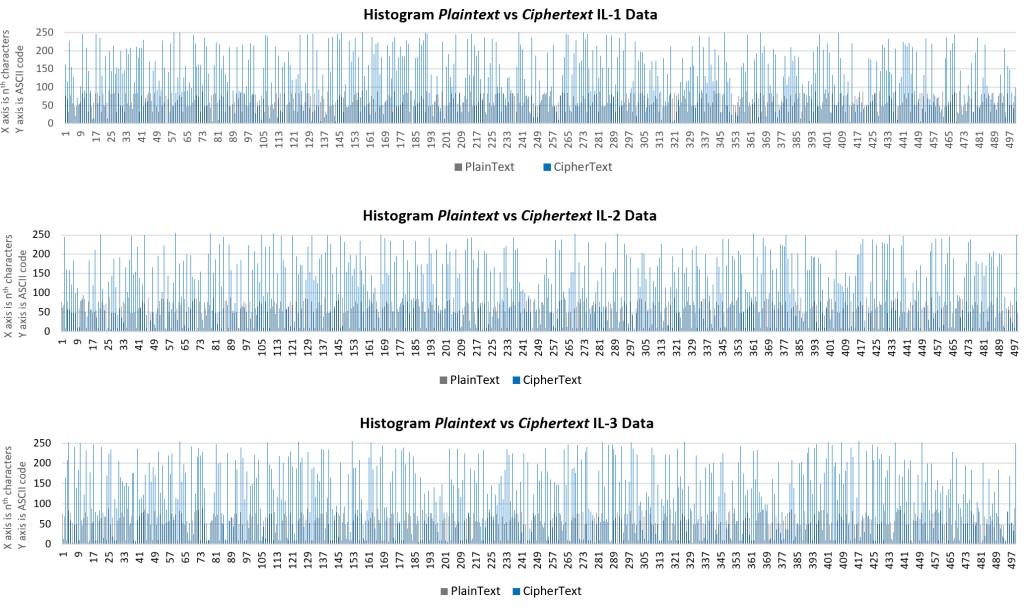

**Figure 13** **Histogram of plaintext and ciphertext IL data for the first 500 characters.**

## Time complexity analysis

Time complexity, or T(n), is measured based on the number of computational steps required to run the algorithm as a function of the input size (n). Calculations are based on multiple operator steps, procedures/functions, control steps, and loops (*Setyaningsih, Wardoyo & Sari, 2020*). For compatibility, the symbols Ci and $\sum$ are used for calculation, as follows: C1 is used to symbolize the assignment; C2 is used to symbolize the number of arithmetic operators used; C3 is used to symbolize built-in procedures/functions such as input, output, or user-defined procedures/functions; C4 is used to symbolize the loop operation; and C5 is used to symbolize the structure of the branching conditions. Finally, $\sum$ is used to represent the number of steps involved in each Ci symbol.

The results of the time complexity calculation for the encryption algorithm (see Appendix 1) are as follows:

T (n) = (12+2n+4n/256) C1 + (5652+35n+7n/256) C2 + (11+n/256) C3 + (1023+7n+n/256) C4 + (3+n/256) C5.

= (12C1 + 5652C2 + 11C3 + 1023C4 + 3C5) + (516C1/256 + 8967C2/256 + nC3/256 + 1793C4/256 + C5/256) n.

T(n) ≈ O(n).

Meanwhile, the results of the time complexity calculation (see Appendix 2) for the decryption algorithm are as follows:

T(n) = (268+515n+3n/256) C1 + (8971+291n+4n/256) C2 + (9+3n+n/256) C3 + (1024+264n+n/256) C4 + (3+n/256) C5.

= (268C1 + 8971C2 + 9C3 + 1024C4 + 3C5) + (131,843C1/256 + 74,500C2/256 + 769C3/256 + 67,585C4/256 + C5/256) n.

T(n) ≈ O(n).

**Table 6  Cryptanalysis solutions for the weaknesses of the Vigenere (Beaufort) cipher.**

| Weaknesses of the Vigenere (Beaufort) and Vernam ciphers | Proposed method (BRC4) |
|---|---|
| - The key must be the same length as plaintext, so the key will be repeated until it is the same length as the plaintext. | ✓ The system generates a key using the keystream generation equation until it has the same length as the plaintext, and thus the key is random and not easily solved. |
| - The keys have to be random. | ✓ The system generates a random initial key for each session, which is always different. |
| - The key must not be reused. | ✓ The system generates a random initial key for each session, which is always different. |
| - The equations used are based on the standard alphabet (modulo 26). | ✓ The system uses modulo 256, resulting in increasingly random values of 256 bytes. |
| - Possible keys are combinations of lowercase letters, with a maximum length of 676 bytes. | ✓ Possible key variations are derived from ASCII code, with a maximum length of 65,536 bytes. |

This analysis shows that the encryption and decryption algorithms may be categorized as having linear complexity, meaning that processing time corresponds positively and linearly with data size. In other words, if the algorithm requires n steps to handle data of n size, it will need 2n steps for data of 2n size.

## Cryptanalysis solutions

Super encryption BRC4 is a proposed method that combines the Beaufort and RC4 ciphers, wherein four symmetric key generators are generated dynamically every session. We must ascertain whether this proposed method can overcome the weaknesses of the Beaufort and RC4 ciphers in isolation. According to *Hughes (2019)*, the Beaufort cipher—as with the Vigenere cipher—has several weaknesses. In their case, Hughes used neither the Vigenère nor the Vernam ciphers, as both needed to meet the same three requirements to comply with Shannon's definition of complete secrecy. According to *Alallayah et al. (2010)*, the Vigenere cipher offers a combination of lowercase alphabetical characters, with a maximum length of 676 (26*26) bytes.

On the other hand, *Fluhrer, Mantin & Shamir (2001)* demonstrated and proved that the RC4 cipher is completely insecure on the Wired Equivalent Privacy (WEP) protocol, with a fixed secret key combined with an initialization vector (IV) modifier for both the 24 and 128-bit modifiers (which are known to encrypt different messages). All of these weaknesses have been anticipated by the proposed method, as shown in Tables 6 and 7.

## CONCLUSIONS

This study shows that a key with a length of 8 characters is less secure from brute force attacks. Only keys of at least 16 characters are secure from brute force attacks. Correlation values of −0.010, 0.006, and 0.001, are produced for IL1, IL2, and IL3, respectively, indicating that the proposed method (BRC4) is better than Beaufort encryption or RC4 encryption in isolation. Meanwhile, information entropy values of 7.84, 7.98, and 7.99 are returned for IL1, IL2, and IL3, respectively, also indicating that that the proposed method (BRC4) is better than Beaufort encryption or RC4 encryption in isolation.

**Table 7  Cryptanalysis solutions for the weaknesses of the RC4 cipher.**

| The weaknesses of the RC4 cipher | Proposed method (BRC4) |
|---|---|
| - The same key tends to be used for all blocks in the same data package. | ✓ The system generates a random initial key (K1), which is different every session, then generates further keys (K2, K3, K4) using a keystream generation equation until it fills up the array (K). There is thus no repetition of keys. |
| - The original RC4 key is limited to 40 bits, and the Initialization Vector (IV) is limited to 24 bits. | ✓ The system generates a random initial key of up to 2,048 bits (256 bytes), or even larger. |
| - RC4 is effective with large keys, and thus attacking a PRGA appears ineffective, even when the most well-known attacks take over $2^{700}$ seconds. It is weak for short keys, as the key is repeated until it fills the array (K) to a full 256 bytes. | ✓ The system generates a random initial key (K1), which is different every session, then generates further keys (K2, K3, K4) using a keystream generation equation until it fills up the array (K). There is thus no repetition of keys. |
| - For each PRGA permutation, the value of the array (S) changes at two locations (at the most). | ✓ The system performs different permutations for every block array, resulting in more varied random values for array blocks. |
| - Permutation is performed only once for all blocks formed, forming a pattern that can be learned by attackers. | ✓ The system performs different permutations for every block array to achieve a random value of 256 bytes. As such, the system performs permutations in the first array block, continues the permutation in the second array block, third, and so on until the last block, and as such it generates a random value that varies for every block array. |
| - It is possible for the same S-Box to be used. The same pseudorandom value may be generated repeatedly, as the user key is repeated to fill the 256-byte array. If a key is used to encrypt 8 bytes, it will thus be repeated 32 times to fill the array. | ✓ If the key used for permutation is only 8 bytes in length, the system uses the keystream generator to generate fill the key byte array without repeating the initial key. |
| - An attacker who manages to obtain multiple ciphertext packets can obtain several bytes of the original message by performing XOR operations on two ciphertext packets. For example, if an attacker successfully intercepts two different encrypted messages that use the same key, the attacker may perform an XOR operation to remove the key sequence's effect. If the attacker manages to uncover the plaintext of one encrypted message, the attacker will easily find other plaintext messages without knowing the correct key sequence. | ✓ To perform encryption, the system generates a random initial key (K1), generates a keystream (K2), generates a key-scheduling algorithm (K3), and generates a pseudo-random key (K4). As such, even if an attacker can obtain the first and the second ciphertext, XOR operations still cannot be used to eliminate the effects of the key sequence, as the initial keys used for the first (K1.1) and second (K1.2) ciphertexts are different. Likewise, K1.1 and K1.2 experience further generation to produce K4.1 and K4.2, which are increasingly different. |

Visual analysis, using a histogram, shows that the distribution of the ciphertext is significantly more varied than the plaintext, and thus it is secure from MITM attacks. Time complexity analysis shows that the proposed method is categorized as linear complexity.

Moreover, previous studies of hybrid security approaches have also used multiple key generation, and almost all have applied an in-depth security system. However, not all have evaluated the performance, keyspace, correlation, information entropy, and time complexity, all of which are significant measures for evaluating data transmission performance. The proposed method uses this metric to measure performance and analyze security based on visual analysis, keyspace, entropy, correlation, and time complexity. Further research should explore other possible super-encryption algorithms for improving the security of data transmission in SCADA systems.

### Funding

This work was supported by doctoral dissertation research from the Directorate General of Higher Education, the Indonesian Ministry of Education and Culture. Eko H. Riyadi received financial support from the Nuclear Energy Regulatory Agency (BAPETEN) for during his Doctoral education in Computer Science at Gadjah Mada University. Universitas Gadjah Mada provided research facilities and financial support through the Doctoral Dissertation Research Grant 3097/UN1.DITLIT/DIT-LIT/PT/2020. The funders had no role in study design, data collection and analysis, decision to publish, or preparation of the manuscript.

### Grant Disclosures

The following grant information was disclosed by the authors:
The Directorate General of Higher Education.
The Indonesian Ministry of Education and Culture.
The Nuclear Energy Regulatory Agency (BAPETEN) for during his Doctoral education in Computer Science at Gadjah Mada University.
Doctoral Dissertation Research:  3097/UN1.DITLIT/DIT-LIT/PT/2020.

### Competing Interests

The authors declare there are no competing interests.

### Author Contributions

- Eko Hadiyono Riyadi conceived and designed the experiments, performed the experiments, analyzed the data, performed the computation work, prepared figures and/or tables, and approved the final draft.
- Agfianto Eko Putra conceived and designed the experiments, analyzed the data, performed the computation work, authored or reviewed drafts of the paper, and approved the final draft.
- Tri Kuntoro Priyambodo conceived and designed the experiments, authored or reviewed drafts of the paper, funding acquisition, and approved the final draft.

### Data Availability

 The codes and raw data are available in the Supplemental Files.

### Supplemental Information

Supplemental information for this article can be found online at http://dx.doi.org/10.7717/peerj-cs.727#supplemental-information.

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
