# Peer review of "Improvement of nuclear facilities DNP3 protocol data transmission security using super encryption BRC4 in SCADA systems"

_PeerJ Computer Science, doi:10.7717/peerj-cs.727_

## Round 0.1 · original submission · Major Revisions

The authors are advised to address all the concerns/comments given by the reviewers in the revised version.

Reviewer 1 ·

Basic reporting

No comment

Experimental design

No comment

Validity of the findings

no comment

Additional comments

The authors have made changes according to the reviewers comments. Now the manuscript is in a good shape and presentation. However, please check again the English. For example, sentence in the last paragraph of Section 2, line 188-189.

As for comparison to the proposed method, as shown in Table 1.

Annotated reviews are not available for download in order to protect the identity of reviewers who chose to remain anonymous.

·

Basic reporting

The authors proposed a method of improving DNP3 security by introducing BRC4 encryption. This combines Beaufort encryption, in which plain text is encrypted by applying a polyalphabetic substitution code based on the Beaufort table by subtracting keys in plain text, and RC4 encryption, a stream cipher with a variable-length key algorithm. Authors contribute to improving the security of data transmission and accelerating key generation. The work by the author is good and the presentation of data is also good.

Experimental design

Experiment design is explained well.

Validity of the findings

For Table 1 Hybrid security approach to data transmission
Which filter used by the author to find the studies for the comparison?

Additional comments

The author has done good work.
They should check grammatical errors in the manuscript.
The quality of images should be improved.

Reviewer 3 ·

Basic reporting

Tests are carried out by key space analysis, correlation coefficient analysis, information entropy analysis, visual analysis, and time complexity analysis.The results show that to secure encryption processes from brute force attacks, a key of at least 16 characters is necessary. IL data correlation values were IL1 = -0.010, IL2 = 0.006, and IL3 = 0.001, respectively, indicating that the proposed method (BRC4) is better than the Beaufort or RC4 methods in isolation. Meanwhile, the information entropy values from IL data are IL1 = 7.84, IL2 = 7.98, and IL3 = 7.99, respectively, likewise indicating that the proposed method is better than the Beaufort or RC4 methods in isolation. Both results also show that the proposed method is secure from MITM attacks. Visual analysis, using a histogram, shows that ciphertext is more significantly distributed than plaintext, and thus secure from MITM attacks. The time complexity analysis results show that the proposed method algorithm is categorized as linear complexity.

Experimental design

Visual analysis, using a histogram, shows that the distribution of the ciphertext is significantly 462more varied than the plaintext, and thus it is secure from MITM attacks. Time complexity 463analysis shows that the proposed method is categorized as linear complexity.

Validity of the findings

no comment

Additional comments

Findings will be useful in the foundations and evaluation of support systems . This version is good in this form but most of the recent references are missing and English of the article is also poor. Comparisons between methods are also missing. Further, please improve these portion, add some significance points of pros and cons of the work and revise the formatting of the references. After the critical revision, this paper can be accepted for publication.

---

## Round 0.2 · accepted · Accept

The reviewers are satisfied with the revision and so do I. I have reviewed the revised version of the manuscript, and I am satisfied with the revision of the manuscript.

Reviewer 1 ·

Basic reporting

No comment

Experimental design

No comment

Validity of the findings

No comment

Additional comments

The authors have addressed the concerns during the first round of review process.

·

Basic reporting

The manuscript by the author still needs improvement, The presentation of data in the manuscript is not good. Lots of points are resolved by the authors in revision.

Experimental design

No comment

Validity of the findings

no comment

Additional comments

Refer to my previous comments

Reviewer 3 ·

Basic reporting

Now, this paper can be accepted for publication.

Experimental design

no comment

Validity of the findings

no comment

Additional comments

no comment